# Potential Risk to Pollinators from Nanotechnology-Based Pesticides

**DOI:** 10.3390/molecules24244458

**Published:** 2019-12-05

**Authors:** Louisa A. Hooven, Priyadarshini Chakrabarti, Bryan J. Harper, Ramesh R. Sagili, Stacey L. Harper

**Affiliations:** 1Department of Horticulture, Oregon State University, 4017 Agriculture and Life Science Building, Corvallis, OR 97331, USA; priyadarshini.chakrabarti@oregonstate.edu; 2Department of Environmental and Molecular Toxicology, Oregon State University, 4017 Agriculture and Life Science Building, Corvallis, OR 97331, USA; Bryan.Harper@oregonstate.edu; 3School of Chemical, Biological and Environmental Engineering, Oregon State University, 116 Johnson Hall, Corvallis, OR 97331, USA

**Keywords:** Nanotechnology-based particles, Pesticides, Bees, Pollinators, Environmental pollution, Exposure potential, Particulate matter

## Abstract

The decline in populations of insect pollinators is a global concern. While multiple factors are implicated, there is uncertainty surrounding the contribution of certain groups of pesticides to losses in wild and managed bees. Nanotechnology-based pesticides (NBPs) are formulations based on multiple particle sizes and types. By packaging active ingredients in engineered particles, NBPs offer many benefits and novel functions, but may also exhibit different properties in the environment when compared with older pesticide formulations. These new properties raise questions about the environmental disposition and fate of NBPs and their exposure to pollinators. Pollinators such as honey bees have evolved structural adaptations to collect pollen, but also inadvertently gather other types of environmental particles which may accumulate in hive materials. Knowledge of the interaction between pollinators, NBPs, and other types of particles is needed to better understand their exposure to pesticides, and essential for characterizing risk from diverse environmental contaminants. The present review discusses the properties, benefits and types of nanotechnology-based pesticides, the propensity of bees to collect such particles and potential impacts on bee pollinators.

## 1. Introduction

Insect pollinators, including both managed and native bees, contribute billions of dollars in value to the agricultural economy [1]. Insects, birds, bats, and other animal species pollinate most of the world’s leading food crops, and there is a trend toward increasing dependence of agriculture on pollination services [2]. Pollinators are integral to plant reproduction and biodiversity, providing ecosystem services that extend far beyond agriculture [3,4]. In the process of foraging from flower to flower, gathering pollen and nectar, pollinators also inadvertently collect diverse environmental contaminants, including pesticides. Complex mixtures of current and legacy use pesticides are routinely found in honey bee tissues and hive matrices [5,6]. Recent unsustainable losses of honey bee colonies, and diminishing populations of native bees around the world, have renewed interest in the potential risk to bees from not only pesticide active ingredients, but their formulations as well [7,8].

Many current use pesticides are formulated from existing active ingredients and other materials such that the formulated product contains particles ranging in size from nanometers to microns [9]. As opposed to a solution of small molecules, particle size and properties enable these formulations to behave like colloidal systems or suspensions in aqueous solution. The size, shape, and surface properties of engineered nanomaterials alter properties of the pesticide product such as solubility, stability and interaction with biological systems [10,11]. Due to diverse composition, there are many ways to describe these formulations [11]. In the present review, we use the term nanotechnology-based pesticides (NBPs) to include the broadest array of particulate formulations.

Since pollinators have structural adaptations to collect and move pollen particles [12,13], and electrostatic forces further assist this interaction, it is plausible that pesticide formulations containing particles could impact pollinator exposure to pesticide active ingredients in ways that were not predicted by the original regulatory assessments of the active ingredient. This potential increased active ingredient exposure could be particularly relevant for honey bee colonies, which may have approximately 50,000 worker bees at peak flowering season [14] functioning as a superorganism, with a well-defined caste system, communication and division of labor. Foragers (Figure 1) make up a large fraction of the colony, and can collect pollen and nectar in a radius of several miles [15]. The continuous collection of pollen and nectar by tens of thousands of individual bees can result in accumulation and concentration of a complex mixture of chemicals in hive matrices. Consequently, the properties of NBP particles, their expected interactions with plant and pest targets and their environmental fate and behavior should be studied to ensure that accurate risks to pollinators are being assessed [16,17,18]. The current lack of this key information makes it difficult to determine whether bees and other pollinators collect and transport these particles to the colony. The present literature review is based on information gleaned from 172 past relevant publications and provides an in-depth characterization of NBPs and an investigation of their potential impacts on bee pollinators.

## 2. Bees as Environmental Sentinels

Bees and other pollinators have evolved many specialized structures to collect and transport pollen [19]. Honey bees trap pollen in the setae (hair-like structures) which nearly cover their entire body, including their eyes [12]. Bees may actively collect pollen, or amass it passively via electrostatic forces [20]. Bees accumulate positive charge as they fly, and induce an image charge as they approach blossoms, which is important in final pollen capture [21] (Figure 2). As bees fly through the air, the friction of flying creates positive charge. Flowers have a negative charge, higher at the tips and edges. As they come close, bees are able to sense the electric field of flowers, and when they come even closer, electrostatic forces facilitate the movement of pollen to the bee’s body. The ability of bees to detect charge as a floral cue of pollination status, has only recently been studied [22,23], and the setae and other organs of bees may be involved in electroreception [24].

While still flying, honey bees clean pollen from their head and mouthparts, transfer it to their hind legs, and push the pollen into the corbicula (pollen basket) for transport back to the colony. Pollinators may inadvertently collect other types of particles. While pollen grains are very large compared to the NBPs discussed here, electrostatic effects are likely responsible for non-pollen particle collection or collection of pollen with particulate contamination. Honey bees collect pollen, together with any contaminants, combine it with nectar and ferment it into bee bread, the protein source for developing larvae [25]. Examining bees collected from the field using scanning electron microscopy, we observed many unidentified non-pollen particles present on their bodies (Figure 3), which were collected by diverse setae across the various honey bee body segments.

Due to their extensive foraging over thousands of acres and their unique morphology, honey bees can pick up minute concentrations of chemicals in the environment. While foraging for pollen, bees collect other broadly dispersed matter which would normally be below the threshold detectable by laboratory analysis [26]. The contaminants are concentrated in hive materials such as pollen, beeswax, nectar, propolis, and the bees themselves, enabling their measurement. Honey bees have been used to detect organic materials such as polychlorinated biphenyls, heavy metals including lead, arsenic, cadmium, and mercury [27,28,29,30], and the dissipation of Cesium-137 years after the Chernobyl accident [31]. It is no surprise that many recent publications have documented multiple pesticides in beeswax, pollen, honey, and bees around the world [5,6,32,33,34,35,36]. In addition to environmental chemicals, colony matrices may contain chemicals such as antibiotics and miticides used by beekeepers to maintain the health of the bees [37]. Consequently honey bees are often considered as important bio indicators of environmental pollution [38,39,40,41,42,43].

## 3. Properties and Benefits of Nanotechnology-Based Pesticides

Nanopesticides are touted as a potential revolution in agriculture, which will facilitate the challenge of meeting increasing global demand for food, fiber, and fuel [44]. The many NBPs already available on the market offer benefits over traditional formulations such as increased solubility, stability and efficacy. Current use pesticide formulations can be labelled as “controlled release”, “improved rain fastness”, “protection from breakdown by UV light”, or “long-lived residual control”, which may reflect how NBPs protect pesticide active ingredients from physical and microbial degradation processes [45]. NBPs may be engineered to release the crop protection agent upon dehydration, changes in pH or temperature, interactions with enzymes or antigens, or other triggers; all enabling increased precision in application. Subsequently, less active ingredient may be needed, and particles can be engineered to increase target deposition [46,47,48]. In addition, NBP formulations can be more environmentally friendly and have reduced risks for applicators by reducing the amount of volatile petroleum solvents traditionally used to deliver poorly soluble active ingredients in many traditional pesticide formulations [49].

Many pesticide active ingredients are hydrophobic, or have other properties that pose barriers to creating aqueous solutions for their application. Nanotechnology-based pesticides package crop protection agents in particles sized to improve solubility or dispersion. Some active ingredients can be crystallized and milled to create nanoscale particles, increasing solubility due to the increased surface area to volume ratio and increasing dissolution [50]. Increased solubility can also be attained by incorporating pesticide active ingredients into or onto the surfaces of amorphous particles [51]. Particulate pesticide formulations may have formulation codes such as capsule suspension (CS), flowable (F), suspension concentrate (SC), soluble concentrate (SL), soluble granules (SG), and others, although the exact nature of pesticide formulations are considered confidential business information and are generally unavailable to the public [52,53].

NBP particles may be coated or suspended in inert ingredients, including surfactants, solvents, emulsifiers, defoamers, stabilizers, anti-microbials, anti-freeze, pigments, buffers, or other materials which endow them with properties absent from the raw active ingredient [54]. These formulation ingredients may themselves be nanoscale particles, which may not improve the delivery or efficacy of the active ingredient, but rather provide UV protection or extend shelf life.

## 4. Size as an Arbitrary Criterion for Regulation of Nano-Agrotechnology

Nanoscale materials or substances can have different or enhanced properties than the same chemical substances with structures at a larger scale and are being developed for various public and commercial applications. These same special properties may cause some of these chemical substances to behave differently than conventional chemicals, and hence require special regulatory review. The International Organization for Standardization (ISO) defines nanoscale as a length range from approximately 1 nm to 100 nm [55]. Particles in the low end of the nanoscale are of particular regulatory interest, because materials may exhibit novel functional or toxicological properties in this size range [37,56,57]. There is ongoing international cooperation to develop approaches and methods to assess risk from objects below the 100 nm threshold [58].

The United States Environmental Protection Agency (EPA) has agreed to regulate novel nanomaterial pesticides, using a working definition of 1 nm–100 nm for nanoscale [59]. Nanosilver is an example of an antimicrobial nanopesticide with particle sizes between 1.5 nm–5 nm (Table 1).

The properties of nanosilver are also based on large surface area relative to larger bulk silver particles, which increases the release of ions at the surface and increases antimicrobial activity [71]. The USEPA granted nanosilver conditional registration as an antimicrobial pesticide, but it has been stalled by legal disputes [72] due to data gaps and uncertainties relating to human exposure and environmental impacts. While nanosilver is a nanomaterial composed of a single substance, and NBPs are often larger heterogeneous particles, NBPs may also have size-related properties. NBPs have not received the same regulatory scrutiny as nanosilver, perhaps because nanosilver is an active ingredient, while NBP particles act as carriers for active ingredients, and NBP particle dimensions are generally above 100 nm. The European Food Safety Authority (EFSA) now recognizes that larger particles may have size related properties that merit regulation, but that there is no agreed-upon definition of the term ‘nanopesticide’ [59].

Another difference from nanosilver is that NBPs are currently released into the environment through agricultural sprays. Other types of anthropogenic activity unintentionally releases diverse particles into the atmosphere. Many industrial processes, including agricultural activities, release dust and other particulate matter (PM), which is a form of air pollution regulated under the US Clean Air Act and EU air quality standards. PM is divided into course and fine particulate matter by aerodynamic diameter of 2.5 µm (PM2.5) [73]. A network of monitoring sites compare PM2.5 and PM10 levels to national standards. These standards are based on research relating inhalation of PM to human respiratory and cardiac health impacts [58,74].

PM and nanoscale dimensions overlap, and nanotoxicology research is providing insights into the impacts of ultrafine particles, PM0.1 [75]. There is already concern that increased use of nanomaterials will contribute to atmospheric PM, and enable novel chemical transformation pathways [73]. While there has been investigation of potential effects of agricultural nanoparticles on soil microfauna and plants [46,76], few studies have been performed to investigate the transfer potential of these particulates to off-target species [77,78]. To date, PM has not been a focus of air pollution effects on arthropods, even those considered critical for pollinating croplands [79] with only a handful of studies looking into the evidence for PM collection by bees [80,81].

We investigated a number of agricultural and residential pesticides using scanning electron microscopy and found particles ranging in size from approximately 50 nm to several microns (Figure 4), suggesting many exceed the ISO size criterion for nanoscale. We also gleaned information from existing literature (Table 1). If airborne, the engineered NBP particles discussed here would predominantly be characterized in the range between PM0.1 and PM2.5 (Table 1). Some examined particles were observed to be very amorphous, or contained multiple types of particles, while others were very homogeneous (Figure 4). For example, Optimate CS is an insecticide with 5.9% gamma cyhalothrin, made by Control Solutions Incorporated (Pasadena, TX, USA) (Figure 4A) where CS formulation code stands for cyhalothrin sold by Syngenta (Basel, Switzerland), capsule suspension. Warrior II with Zeon Technology is an insecticide with 22.8% lambda- and the label also notes it as a capsule suspension (Figure 4B).

Natria is a Ready-to-Use (RTU) insecticide containing pyrethrins and sulfur manufactured by Bayer (Leverkusen, Germany) (Figure 4C). Rovral 4F is a fungicide containing iprodione (Figure 4D) sold by FMC Corporation (Philadelphia, PA, USA) where 4F stands for 4 lbs/gallon active ingredient in a flowable formulation. Tourismo is an insecticide/insect growth regulator (Figure 4E) containing 12.5% flubendiamide and 25.0% buprofezin sold by Nichino America, Inc. (Wilmington, DE, USA), and does not list a formulation code. Protocol is a fungicide containing 23.7% thiophanate methyl and 7.1% propiconazole (Figure 4F), is sold by Loveland Products (Salem, MA, USA), and no formulation code is listed on the label. Bravo Weatherstik is a fungicide containing 54% chlorothalonil (Figure 4G) made by Syngenta (Basel, Switzerland), and the label notes it is a suspension concentrate (SC).

Intrepid 2F is an insect growth regulator containing 22.6% methoxyfenozide (Figure 4H) made by Dow AgroSciences (Indianapolis, IN, USA) where 2F indicates 2lbs active ingredient/gallon in a flowable formulation. Bayer Advanced All-In-One Lawn Weed & Crabgrass Killer (Figure 4I,J) is a RTU herbicide containing three active ingredients: 2,4-D, quinclorac, and dicamba. Small particles were observed which appeared to be separate as well as mixed with larger particles (Figure 4I,J). Tempo SC Ultra is an insecticide containing 11.8% beta-cyfluthrin (Figure 4K) made by Bayer (Leverkusen, Germany). The SC formulation code stands for soluble concentrate, and it contains sets of particles that appear distinctly different from one another. Safari 2SG is an insecticide containing 20% Dinotefuran (Figure 4L) sold by Valent (Walnut Creek, CA, USA) where SG is an abbreviation for soluble granules.

The composition and nature of pesticide formulations we observed in Figure 4 are proprietary business information, and we are unable to determine the composition of the particles beyond their inclusion of the pesticide active ingredient [53]. However, we can learn a great deal about the various types of NBP formulations currently in the marketplace (Figure 4). Our literature review found that particles in pesticide formulations might have layers, pores, coatings, or other structures in the nanoscale, which greatly increase surface area for interaction with pesticide active ingredients. Some NBPs are based on nano or micron sized particles and emulsions of existing pesticide active ingredients, coated or encapsulated by a nanolayer of another substance, or adsorbed onto, or entrapped within a polymeric particle [82]. For example, encapsulated formulations provide a polymer shell as a water dispersible carrier enveloping a core or reservoir including the active ingredient. The capsule may be designed to burst open and release its contents after application. The shell can be single or multiple layers of synthetic, naturally occurring, or biodegradable polymers [83], and can alter environmental properties and toxicity [84] (Figure 5A). Monolithic Polymeric Spheres involve homogenous aggregation of the active ingredients with matrix materials, which are then processed into spheres to aid release of the active ingredient over time [85] (Figure 5B). Solid lipid nanoparticles are a similar approach [86]. Polymers and hydrogels are natural or synthetic polymers cross-linked to form a gel, to which pesticide active ingredient is attached [87,88] (Figure 5C).

For suspension NBPs, crystallized active ingredients are milled to sub-micron size in the presence of compounds that coat the crystals. The size and coating of the crystals allows them to be suspended in aqueous solutions [54,89] (Figure 5D). Porous silica are silicates and aluminosilicates to which organic moieties can be attached [90]; whereas zeolites are diverse porous, crystalline aluminosilicates of natural or synthetic origin and considered as an inorganic polymer which can act as a slow release carrier for pesticides [33] (Figure 5E). Natural clays are composed of layered silicate sheets or platelets, which provide surfaces for pesticide adsorption [91]. Organoclays and clay composites contain cations, polymers, or other structures between the layers [91] (Figure 5F). Layered double hydroxides exhibit diverse, positively charged layer structure balanced with anions in the interlayer [92] (Figure 5F). The use of clay particles to protect, stabilize, and slowly release crop protection agents has been utilized for decades [93]. Metal nanoparticles are particles under 100 nm in diameter, which have functional activity without addition of a traditional pesticide active ingredient. While there is interest in the use of metal nanoparticles in agriculture, safety and regulatory concerns remain as noted above with nanosilver [94,95] (Figure 5G). Nanoemulsions are water-based systems containing nano sized hydrophobic droplets which can be used as colloidal carriers for pesticides [82,96] (Figure 5H). In addition to the examples illustrated in Figure 5, many other ingredients, and hybrid particles have been proposed or are currently being used for delivery of crop protection agents.

Studies by Kah et al. [9,97] have proposed that nanopesticides be defined as formulations that (a) intentionally include entities in the nanometer size range (including entities up to 1000 nm), or (b) are designated with a “nano” prefix (e.g., nanohybrid, nanocomposite), and/or (c) is claimed to have novel properties associated with the small size [9,97]. When particles above 100 nm are included, many currently used pesticides fit Kah’s definition [9,97] (Figure 4 and Figure 5). Regardless of the size or composition of the particles in formulated pesticide products, they potentially alter the fate of the active ingredients, which creates uncertainty about environmental fate and non-target exposure and toxicity assessments that fail to include colloidal or particulate formulation risks.

## 5. Risks Posed by Particulate Contaminants on Pollinators

In the 1970s, a microencapsulated methyl parathion product called Penncap M was introduced for insect pest control. Methyl parathion is known to be highly toxic to bees, and the capsules, approximately the size of pollen grains (Table 1), adhered electrostatically to bees, and persisted longer in the field and in pollen stored in the hive than expected from the active ingredient alone [98]. Beekeepers experienced significant colony losses due to exposure of their colonies to this organophosphate insecticide, before its use was cancelled in the US in 2011. These losses reiterated the admonition to avoid exposure of bees to encapsulated and dust pesticide formulations [99]. Current encapsulated NBP pesticide formulations are orders of magnitude smaller in particle size (Table 1, Figure 4 and Figure 5).

In recent years, beekeepers have been contending with new honey bee colony losses associated with fugitive dust from planting insecticide-coated seeds. In this scenario, corn and soy seeds were pre-treated with a coating containing neonicotinoids such as clothianidin, thiamethoxam, or imidacloprid. The seedling draws up the systemic insecticide as it grows, protecting it from insect pests. At first, this seemed like a clever means of applying pesticides right where they were needed, with little expected exposure to bees. However, the coated seeds tended to stick to each other in the vacuum seed planter, necessitating lubrication with talc or graphite powder. The seed coating was abraded during the planting process, and bound to the lubricant powder as it passed through the mechanized planter. The dust was released from planter exhaust, and blew across the landscape to trees, woody plants, and nearby fields where bees forage. This fugitive dust was associated with high mortality for honey bee colonies [64,100]. The particles of talc and graphite ranged as low as 230 nm (Table 1), and honey bee colonies located several miles from the planting site were affected. Low concentrations of neonicotinoids may also remain in surface soils, which may drift after wind erosion [36,101].

Similar scenarios have been described in Europe, the U.S., and Canada. While these pesticide-associated particles were generated unintentionally, this is a cautionary tale illustrating the urgency of investigating the environmental fate of intentionally engineered NBP particles. The discovery of negative impacts of fugitive neonicotinoid laden dust on bees has led to serious efforts to develop alternative lubricants and best practices to mitigate this problem [102].

It is unknown whether intentionally engineered NBP particles contribute to fugitive dust or atmospheric PM. Fugitive dust in the Central Valley of California regularly exceeds particulate matter regulatory limits [103]. In California’s San Joaquin Valley, persistent pesticides may be a component of fugitive dust [104]. Balancing agricultural production and concomitant disruption of surface soils with protection of human health has been identified as an important concern [105]. Potential effects on ecological systems, including pollination networks, should also be a research priority.

In addition to pesticides, other types of PM may pose hazards for bees. Soot particles are formed during the incomplete combustion of diesel fuel, and recent work indicates that diesel exhaust interferes with the ability of bees to receive olfactory cues from flowers [62,63], and air pollution has been found to disrupt floral scents and increase foraging times [106]. Volcanic ash, a natural source of nanoparticles, may disrupt plant/pollinator interactions [107]. Bees can also inadvertently collect biological particles, including airborne bacteria [108], viruses [109], fungal spores [110], transmit plant pathogens [111], and have been investigated as potential delivery agents of biocontrol agents [112]. Clearly, bees and likely other pollinators that collect and consume pollen inadvertently, collect particles in the air and the terrestrial environment which may increase their exposure to toxic materials [80].

## 6. NBP Uncertainties for Pollinators

The size of NBPs may influence toxicity in other organisms, including aquatic systems [113]. Preliminary laboratory studies have also indicated that nanosilver may have potential adverse effects on honey bees [60]. However, to understand how NBPs may affect bees, it is necessary to consider how pesticide particles are applied in the field, and how that intersects with particle collecting abilities of pollinators. Pesticides are often applied using electrostatic spray nozzles, resulting in a cloud of charged droplets. The charge to mass ratio of the droplets is very high, and electrostatic forces drive deposition on the target. There are many variables such as the spray device and interaction with the surrounding air, that impact final deposition on a target surface [114]. Coatings to increase electrostatic properties of particles increase pesticide availability to mosquitoes [115] and electrostatic particle approaches are being used to address other pest insects [116,117,118]. Charged droplets and particles are likely to have an even greater affinity for the pollen-attracting features of pollinators.

Label precautions prohibit spraying bee-hazardous pesticides during pollination under most conditions, but sprays may drift many miles from the application site [119], and non-target plants may expose bees to pesticides [120]. Newer electrostatic spray methods reduce, but do not eliminate drift. Although there is a lack of quantitative exposure data for non-target sites and organisms [121], there is continuous improvement in modelling drift, with a focus on droplet size [122]. Particles within formulations are not expected to appreciably affect the formation of droplets or drift potential [123]. However, the drift of particles after water has evaporated from the droplets in the air, or aeolian effects on particulate residues settled on foliage or in soil, has not been explored to our knowledge, and may be more appropriately modelled by approaches used to describe atmospheric dispersion of particulate matter. Interestingly, studies of pesticides in the atmosphere find that current use pesticides are associated with mineral-based particles [124,125,126]. NBPs may be made of clay and other minerals, and pesticides may also sorb to soil particles. Some pesticides may be found at sites distant from their possible application, at odds with known properties of the active ingredient [127]. Similarly, it is currently difficult to account for the diversity of pesticides found within colony matrices [128]. If NBP drift contributes to the atmospheric pool of PM, and such particles are accumulated by bees as they fly through the air, it could help explain the complex mixtures of pesticides found in hive materials.

The electrostatic forces involved in pollen transferring from flower anthers to a bee, and from a bee to stigma, have been harnessed in devices used for mechanical pollination [129]. It should come as no surprise that pollen attracts PM particles [67], and we have found particles adhering to almond pollen collected from the field, which resemble NBPs (Figure 6A). Leaves and petals have complex morphology, surface topology and chemistry [130], which may affect the deposition and behaviour of particles (Figure 6B), and the transfer of particles to pollinators. The tips and edges of leaves and flowers are expected to possess the greatest magnitude of negative charge, although the electric fields surrounding plants can change in polarity under unstable weather conditions [21].

NBPs may reach target sites at higher doses compared to conventional formulations [131]. Since bees interact with plants at the projecting reproductive organs of flowers, this could influence the amount and type of NBPs they are exposed to. In addition to passive accumulation, bees can actively gather pollen [12]. Passive particle accumulation and active pollen collection by bees could increase their exposure to environmental contaminants, including NBPs. The EPA’s Bee Rex model, which estimates contact and ingestion exposures to individual bees based on application rate, does not currently consider potential variability in behaviour of NBP particles which could alter exposure [132].

Once pesticide droplets have evaporated, there is little public information available indicating how tightly or efficiently NBP particles or residues adhere to foliage and flowers. This creates uncertainty whether NBP particles can transfer readily from foliage to pollinators, and whether this varies over time or other conditions. Controlled release, while extending the period of efficacy for controlling pest organisms, also implies longer residual toxicity for non-target organisms. The length of residual toxicity is an important variable in agricultural systems, where the arrival and departure of managed pollinators must be coordinated with pesticide applications [99].

Direct contact with some insecticides can kill bees in the field. Controlled release NBP formulations could protect bees from immediate exposure, but enable them to transport particles, leading to delayed risk to the colony. Complex mixtures of pesticides are commonly found in bees, nectar and honey, pollen, bee bread (Figure 7) and beeswax [133]. Pesticide residues transported with pollen are of increasing concern, due to the importance of this sole protein source in larval development [134]. The final instars of honey bee larvae are fed small quantities of pollen directly, and also the nurse bees consume significant quantities of pollen to produce brood food for the developing larvae [135]. Due to long-range foraging of bees, the majority of hive materials, even when the hives are near organically managed crops or remote areas are contaminated with pesticides. The evolution of pesticide active ingredients, improved sample preparation, and analysis methods would make it difficult to make historical comparisons [136]. Laboratory and field studies are needed to investigate whether NBP formulations increase the accumulation of any given pesticide in hive matrices.

Residues in hive materials are a source of uncertainty in pesticide hazards for bees [128]. The properties of pesticide active ingredients such as hydrolysis and photolysis in soil, solubility in water, vapor pressure, n-octanol-water partition coefficient (KOW), Henry’s Law Constant, and dissociation constant in water (pKa or pKb), are used to model environmental fate and predict dissipation processes [128,137]. The hydrophobic nature of many pesticide active ingredients can be factored into models of pesticide accumulation and persistence within hive matrices [138,139]. Due to their hydrophobic nature, many pesticides accumulate in bees wax. Contaminated beeswax may adversely affect bees [140,141], although risk assessors assume low risk of pesticide exposure from wax to bees [142].

NBPs are frequently engineered to be water soluble, and may pose problems in modelling toxico-kinetics of hydrophobic compounds [143]. If transported into the colony, hydrophobic active ingredients could potentially accumulate in the higher exposure risk aqueous compartments such as nectar and honey, or stored in pollen and bee bread, or the bees themselves. This could alter the pesticide exposure to larvae, or particularly in the case of fungicides, affect the microorganisms which are essential for fermenting pollen to more nutritious beebread [134,144,145,146,147,148,149]. Active ingredients protected within an NBP particle could be protected from metabolism by bees and the microbiota, which ferment pollen into bee bread. Depending on the nutritional needs of the colony, contaminated food can be consumed immediately, gradually, or much later [68]. Novel properties of NBPs could make prediction of pesticide fate within hive materials even more difficult, and promote delayed and chronic pesticide exposures to developing bees.

Moreover, our understanding of pesticide drift is based on the effects of wind and weather on droplets during application, and the evaporation of volatile pesticides. The forces that potentially influence fate, movement, and physical and trophic transfer of NBP particles in terrestrial environments remain largely unexplored [78]. This knowledge is particularly important for understanding how bees and other pollinators are exposed to various environmental contaminants.

## 7. Implications: The Need to Consider NBP Formulations in Risk Assessment

Colony losses associated with pesticide use are often mysterious, such as the 80,000 or more honey bee colonies that were negatively impacted during California almond pollination in 2014, for which a cause was only recently tentatively attributed [150]. The accumulation of certain pesticides in colonies has been linked to Colony Collapse Disorder, although no cause and effect relationship has been verified [151,152]. Suspicions persist about the role of neonicotinoids in pollinator losses [153]. Multiple wild pollinators such as various bumble bee species, Monarchs, and other butterflies are struggling, and pesticides may exacerbate the effects of other stressors [154,155]. These lingering questions underscore the need to better understand pesticide exposure to bees and other pollinators. NBPs can potentially change the route, amount and residual toxicity to non-target organisms, and these properties must be considered as we struggle to clarify these uncertainties.

Conversely, NBPs may be reducing risk for pollinators, by decreasing drift, use of solvents, and the amount of pesticides needed for effective use. By extending the half-life of less stable compounds, they could enable delivery of less toxic, and more targeted approaches such as RNAi [114,156]. They could be engineered with protection of pollinators in mind, with the release of active ingredients triggered by interaction only with specific pests and pathogens [157]. In particular, these potential attributes of NBPs could be leveraged to address the scourge of honeybees, the ectoparasitic mite Varroa destructor [158]. In addition to feeding on the hemolymph and fat body tissues of adult and developing bees, the mite carries multiple viruses, and is arguably a far greater threat to honey bees than pesticides [159]. Without more detailed information about NBP formulations, uncertainty and unpredictability remains regarding their exposure or potential benefits to pollinators.

It is imperative that the agricultural and scientific communities make every effort to investigate not only potential benefits, but also potential risks from new pesticide formulation technologies [160,161,162], and NBPs have long since entered the marketplace. Although some of the concern with NBPs is due to the size of the particles, the lack of information on the particles being used in formulated products coupled with minimal knowledge on the fate and behavior of these formulations is currently hampering the ability to conduct accurate risk assessments for the use of these pesticides around pollinators. Studies investigating the fate of particulate pesticides, especially those with features in the 1–100 nm nanoscale, which have the highest probability of unique properties, will help improve protection for pollinators by allowing more informed risk management decisions on their use.

In contrast to a size-based criterion for nanopesticide regulation, the nonbinding recommendations to evaluate whether an FDA-regulated product involves the application of nanotechnology acknowledge that particles above 100 nm may also exhibit size-dependent properties [163]. This takes into account physical and chemical properties and biological behavior of particles above 100 nm that are “relevant to evaluations of safety, effectiveness, performance, quality, public health impact, or regulatory status”. This guidance is in line with Kah’s definition, and suggests that any additional regulatory review of nanotechnology should be based on potential risk.

We are currently applying 5.6 billion pounds of insecticides, fungicides, herbicides, and other pesticides into agricultural environments around the world [43,164], some percentage of which are NBPs. In fact, agrochemical pollutants are considered one of the important factors contributing to global declines in general insect populations [165] as well as in bee pollinators [166,167,168]. Threats and extinctions of insects [169], pollinators, and ecological interaction network links are of increasing concern [169,170]. There is a critical need for researchers and regulators to better understand the complex processes involved in pesticide exposure to individual bees and accumulation within the hive [17], including the environmental behavior of NBP particles in agricultural settings. Such formulations are also used in residential settings, which could similarly impact urban pollinators [171]. Unique evolutionary structural adaptations to accumulate minute particles, such as the general feature of electrostatic pollen collection [172] can predispose all pollinators to a higher risk of exposure. Honey bees are an ideal model organism to investigate how various chemical substances move through the troposphere by hitching a ride on synthetic and naturally occurring particles.

## Figures and Tables

**Figure 1 molecules-24-04458-f001:**
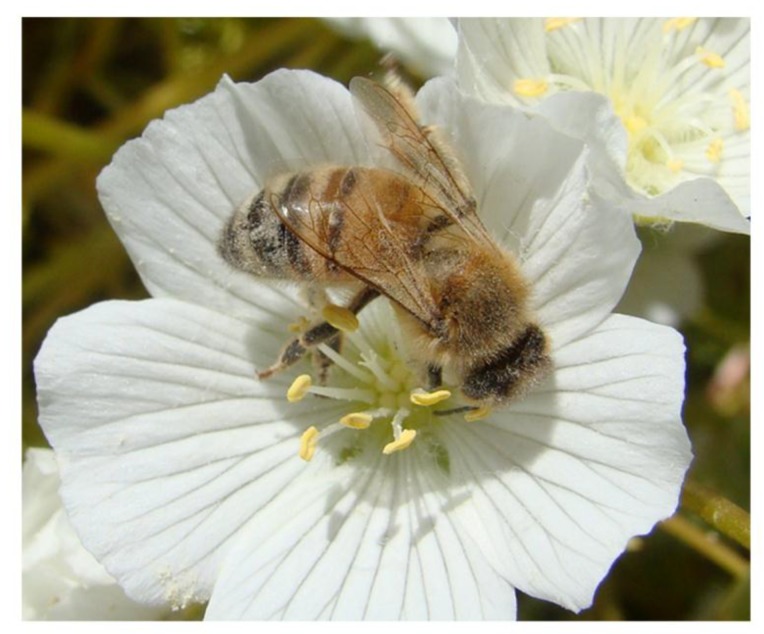
A honey bee (*Apis mellifera*) forager, foraging on meadowfoam (*Limnanthese alba*), a specialty oilseed crop grown in the Willamette Valley of Oregon, USA. Upon close inspection, pollen grains can be seen clinging to the setae (specialized hair-like structures) of the bee.

**Figure 2 molecules-24-04458-f002:**
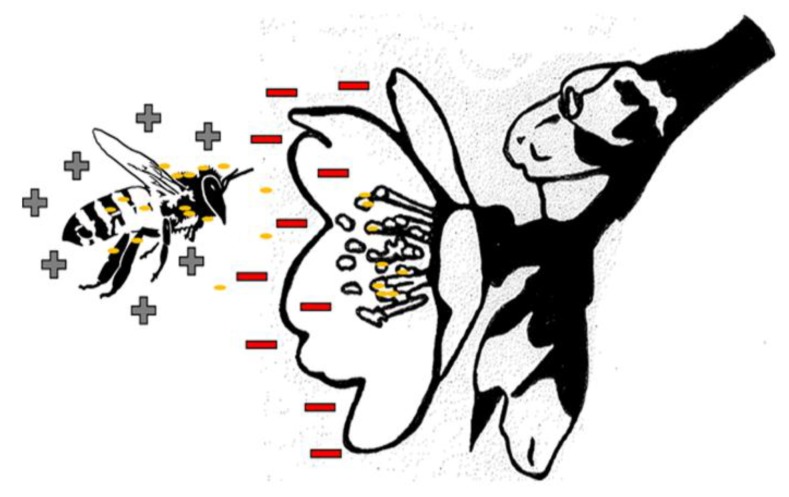
Electrostatic processes contribute to the movement of pollen from flowers to bees. Specialized structures such as hairs contribute to the bees’ ability to accumulate pollen. Bees also actively collect pollen with their forelegs and mandibles.

**Figure 3 molecules-24-04458-f003:**
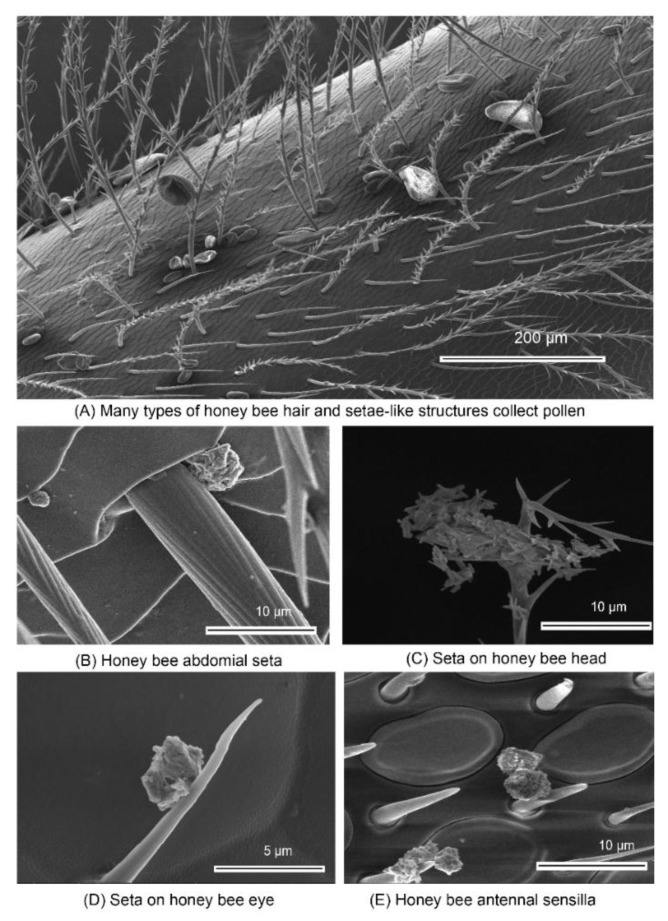
Honey bees have specialized structures that collect pollen and other particles. The hair-like structures trap pollen (**A**), and upon greater magnification much smaller, unidentified particles can also be observed (**B**–**E**) collected by other setae and antennal sensilla. Foraging honey bees (*Apis mellifera*) were obtained from field. Scale as indicated in the figures. Imaging was done at the Oregon State University Electron Microscopy Center using Quanta 600 FEG Scanning Electron Microscope (Thermo Fisher Scientific, Hillsboro, OR, USA).

**Figure 4 molecules-24-04458-f004:**
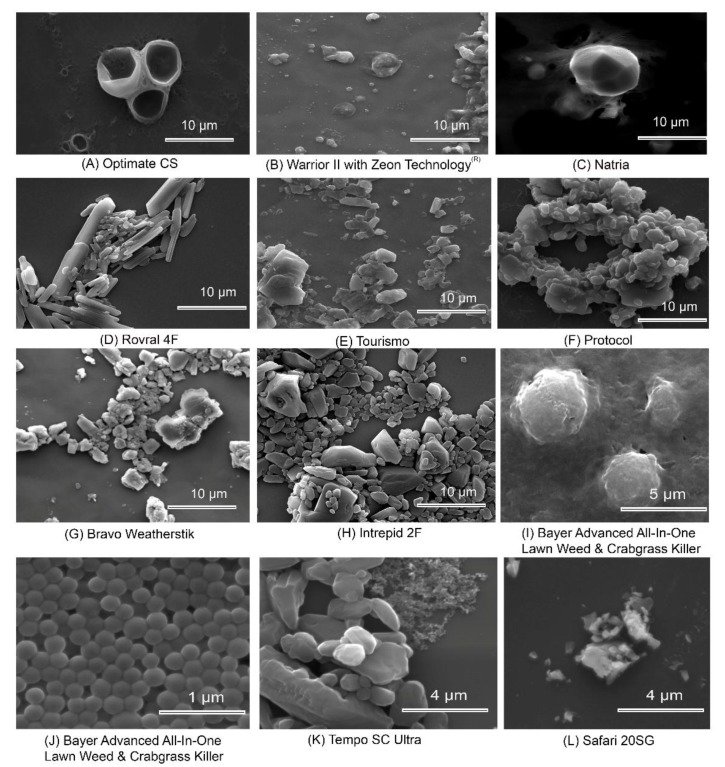
Scanning electron photographs of various pesticides and pesticide formulations: (**A**) Optimate CS; (**B**) Warrior II with Zeon Technology; (**C**) Natria; (**D**) Rovral 4F; (**E**) Tourismo; (**F**) Protocol; (**G**) Bravo Weatherstik; (**H**) Intrepid 2F; (**I**) and (**J**) Bayer Advanced All-In-One Lawn Weed & Crabgrass Killer. A larger structure appears to enclose small particles (**I**); the smaller particles (**J**) were also observed separately; (**K**) Tempo SC Ultra and (**L**) Safari 2SG. Scale has been indicated in the figure plate. Commercially available pesticides in Figure 4 were obtained from local retail establishments, agricultural chemical suppliers, or distributors. Samples were serially diluted with water and pipetted onto a silica substrate, in order to best visualize individual particles. Imaging was done at the Oregon State University Electron Microscopy Center using Quanta 600 FEG Scanning Electron Microscope (Thermo Fisher Scientific, Hillsboro, OR, USA).

**Figure 5 molecules-24-04458-f005:**
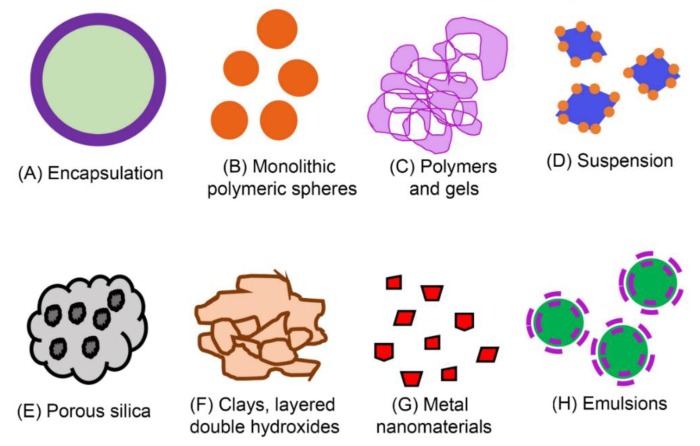
Examples of nanotechnology-based pesticide (NBP) formulations. Many NBPs are assembled from one or more pesticide active ingredients and multiple types of macromolecular matrices. (**A**). Encapsulated formulations; (**B**). Monolithic Polymeric Spheres; (**C**). Polymers and hydrogels; (**D**). Suspension; (**E**). Porous silica and zeolites; (**F**). Clays, layered double hydroxides; (**G**). Metal nanoparticles; (**H**). Emulsions.

**Figure 6 molecules-24-04458-f006:**
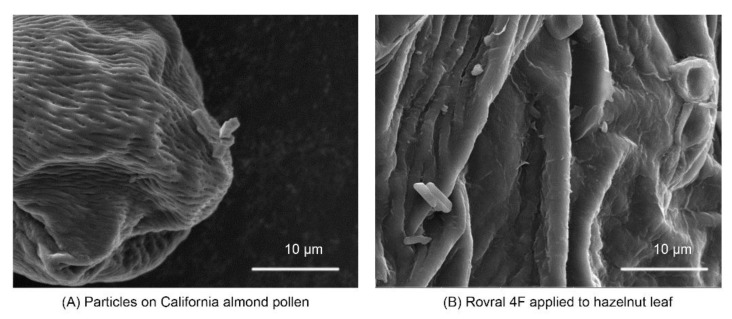
Figure depicts how particles may associate with pollens. (**A**) Particle as found on pollen collected in California almond orchards during pollination. (**B**) For comparison, Rovral 4F was applied to a hazelnut leaf using a spray bottle. Almond pollen was collected from orchards in California. Images were acquired on a Quanta 600 FEG Scanning Electron Microscope (Thermo Fisher Scientific, Hillsboro, OR, USA). Scale is 10 μm.

**Figure 7 molecules-24-04458-f007:**
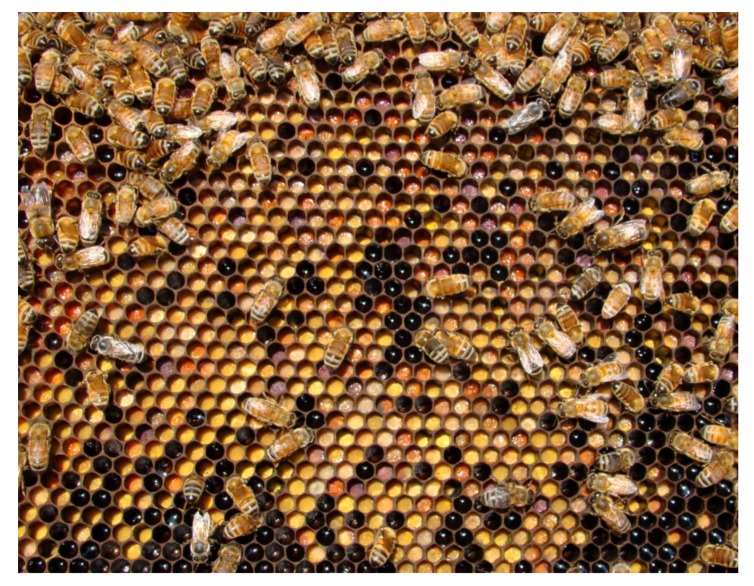
Figure depicts how beebread looks like on a frame. Pollen collected by worker bees is mixed with nectar and fermented by multiple microorganisms into beebread, an essential protein source for developing bee larvae. Nectar is stored as honey, which sustains the colony over the winter. If nanotechnology-based pesticide (NBP) formulations increase accumulation or persistence of pesticides in these compartments, delayed toxicity could result.

**Table 1 molecules-24-04458-t001:** Interactions between nano and micro sized particles and bees.

Particle Classification	Particle Examples	Range	Effects on Bees
Nano (definition varies) PM0.1 <100 nm ultrafine particles	Nanosilver	1.5–5 nm	Decreased *Nosema* spores, variable effects on longevity [60], biocides against American foulbrood pathogens [61].
	Diesel exhaust	7.5–1000 nm	Affects learning and stress response [62], degrades floral odors [63].
	Nanopesticide particles approximate lower range	≈50 nm	Whether NBP size or properties affect exposure or toxicity to bees remains to be investigated
	Fugitive dust from seed planting	230 nm–32 μm	Associated with bee mortality [64,65,66]. Particle fraction under 1 μm contains more active ingredient
PM2.5 <2500 nm fine particles	Nanopesticide particles approximate upper range	≈1–10 μm	Whether NBP size or properties affect exposure or toxicity to bees remains to be investigated
PM10 <10,000 nm coarse particles	Pollen	6–100 μm	Source of protein [25]. Can be a vector of contaminants and smaller particles into the hive [67]
	Microencapsulated methyl parathion (PENNCAP-M)	30–50 μm	Colony mortality, storage in pollen [68,69,70]

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
