# Peer review of "Potential Risk to Pollinators from Nanotechnology-Based Pesticides"

_molecules, 2019, doi:10.3390/molecules24244458_

Round 1

Reviewer 1 Report

A fantastic paper - to be honest when I opened I thought this was going to near kill me reading it given the topic, but it was very well written and very accessible (within reason obviously). I think this is a highly significant and controversial issue. The impact of formulations and NBPs in altering toxic effects is potentially huge and sits outside the regulatory framework which focuses on the active ingredient. A valuable paper helping to define risks in this emerging area for pollinators. Great stuff. I have comments, but nothing major, more suggestions.

Ben woodcock

Abstract

L20 would ‘risk’ be better than ‘exposure’

L21: I would be explicit here and state that one of those potential other particles are NBPs

L22-24: I found this sentence a bit confusing . Firstly what ‘other types of particles’, secondly is this risk of exposure to pesticides that are different to the pesticides within NBPs, finally I would end it as ‘ better understand the risks they post as they interact with other environmental contaminants’.

Keywords

These seam fine

Introduction

L35: I am not sure how true this is (or at least its qualified). Are you talking about an increased area of pollinated crops, as I know for many partially pollinated crops many new varieties have reduced pollinator dependence, not more.

L43: You need a reference her e for declines of wild pollinators. See potentially Powney, G.D., Carvell, C., Edwards, M., Morris, R.K.A., Roy, H.E., Woodcock, B.A., Isaac, N.J.B., 2019. Widespread losses of pollinating insects in Britain. Nature Communications 10, 1018.

L63: I think you need to specify that this could occur in a manner that is fundamentally different to what original regulatory assessments predicted for that product.

L68: Potentially see also Milner, A.M., Boyd, I.L., 2017. Toward pesticidovigilance. Science 357, 1232-1234. Which shows a similar ting (for neonicotinoids).

L75-77: So I guess the obvious thing is to ask if ther is scope for something more formal than just a literature review – metanalysis for example. It may eb the case that current data is not expansive enough for this once you take into account the needs for some level of standardisation.   Possibly need to justify why you take this approach and not something like a meta-anlaysis.

L84: So given how hairy bees are relatie to other pollinating insects does this mean they are at a greater than expected risk (e.g. say compared to flies or beetles)

L97: So this specifically relates to honeybees (i.e. you say bees here). I know you specify it earlier but probably need to always use honeybees. Note many other wild bees do not have corbicula on the back legs (its only really bombus and apis).

L105-108 (by picture) – I am guessing this is unidentified particle. Make this clear that its not necessarily a NBP.

L121: Where do you get this from?   As a back of the envelope calculation if you assume bees forage around 10 km or less from hives this is 7853 ha. If its justified this is ok, but if its of the top of your head I would be careful with this.

L130: also see Woodcock, B.A., Ridding, L., Freeman, S.N., Pereira, M.G., Sleep, D., Redhead, J., Aston, D., Carreck, N.L., Shore, R.F., Bullock, J.M., Heard, M.S., Pywell, R.F., 2018. Neonicotinoid residues in UK honey despite European Union moratorium. PLoS ONE 13, e0189681.

L157-159: I think this is a really important point – I totally agree that when it comes to applying pesticides commercial confidentiality should not be allowed.

L172-174: I think this in some form should go in the abstract – is their space or are you limited by word count.

L177: It would be good if you could extend information on regulation (in brief) to other zones – ie..whats happening in Europe, Asia etc. Don’t need to kill yourself here, but make this more than just about the US.

L190-191: Again I think this is such an important point, the lack of clarity on how formulations including NBPs should be treated from a regulatory perspective given that while not an active ingredient they fundamentally alter properties of the pesticides.  

L222 Great figure. One that would be very cool would be to have the same pesticide but for different formulations (I am not clear if this figure may have this a t least in some of the panes).

L235-255: I wonder whether some of this detail could go in an appendix, to help with the flow of the paper. Maybe, not sure on this.

L256: I find this deeply worrying

L300-303: Should this definition not go earlier on in the review?

L310: mentioned this earlier but your scope seems to vary. Is this focused on honeybees specifically, bees in general or pollinators (which can be a much wider taxonomic group to include for example flies and butterflies). The reason I think you could perhaps focus on bees is your emphasis on electrostatic forces and how hairs exasperate these. Certainly this would seem to be relevant for most bees (note while they don’t all have corbicula, they do tend to have pollen storage structures of one type or another), but may be less pertinent for say anthomyiidae flies (note you may want to consider some hoverflies as due to their mimicry they can be hairy).

L320-321: Add a reference for this – it’s a lot of detail in the paragraph, best to add some citations. E.g. Cresswell, J.E., 2011. A meta-analysis of experiments testing the effects of a neonicotinoid insecticide (imidacloprid) on honey bees. Ecotoxicology 20, 149-157. But frankly there billions of NI references to pick from.

L333: Again see PLoS ONE 13, e0189681.

L348-357: I wonder whether this warrants a separate section – its short I know but its focus on non-pesticide risks associated with nano-particles is distinct in a way.

L371: So these tend to be orientate towards honeybees, other wild bee species can have longer flight periods throughout the day and as such may still be at risk even where these spraying time precautions are applied.

411: So one thing that may be relevant (sorry if you consider it later on) is the effect of the part of NBPs on the persistence of the pesticides activity – I am thinking here of sub lethal effects most widely linked with neonicotinoids as seed treatments and how long term low level exposure risks took a long time to be appreciated. If this changes dramatically with the use of NBPs this could create new problems for existing pesticides. This is especially true given the problems with assessing LC50s under lab conditions – i.e. the control bees tend to die quickly so even long term chronic exposure assessments rarely extend beyond 10-12 days.

L430: Another potentials consideration is if NBPs alter solubility in oils / water – i.e. could end up in wax where this was not an issue before – or vice versa

L449 – ok great you do cover this point.

Author Response

A fantastic paper - to be honest when I opened I thought this was going to near kill me reading it given the topic, but it was very well written and very accessible (within reason obviously). I think this is a highly significant and controversial issue. The impact of formulations and NBPs in altering toxic effects is potentially huge and sits outside the regulatory framework which focuses on the active ingredient. A valuable paper helping to define risks in this emerging area for pollinators. Great stuff. I have comments, but nothing major, more suggestions.

Ben woodcock

Response: The authors are very appreciative for your comments. This is an excellent summary of the points we are trying to make. We have carefully considered each of your comments and incorporated most of them in the revised manuscript text.

Abstract

L20 would ‘risk’ be better than ‘exposure’

Response: We appreciate the reviewer’s concerns. The main question is whether formulating pesticides as particles enables bees to pick them up more easily, so the authors intend to allude to “exposure” in the manuscript context.

L21: I would be explicit here and state that one of those potential other particles are NBPs

Response: We appreciate the reviewer’s suggestions. We think this may be redundant since it is mentioned in the next sentence. We also talk about particles other than NBPs in the paper, so we are referring to particles in the general sense here. We hence edited L23 in the track changed version of the revised manuscript as per the reviewer’s suggestions, trying to avoid redundancy at the same time.

L22-24: I found this sentence a bit confusing . Firstly what ‘other types of particles’, secondly is this risk of exposure to pesticides that are different to the pesticides within NBPs, finally I would end it as ‘ better understand the risks they post as they interact with other environmental contaminants’.

Response: Although the paper is focused on NBP, other types of environmental particles are discussed. We are not enumerating them here because it could distract from the main point of the paper. Based on the reviewer’s suggestions, we edited L25-26 in the track changed version of the revised manuscript.

Keywords

These seem fine

Introduction

L35: I am not sure how true this is (or at least its qualified). Are you talking about an increased area of pollinated crops, as I know for many partially pollinated crops many new varieties have reduced pollinator dependence, not more.

Response: We appreciate the reviewer’s concerns. The reviewer righty notes that there are many ways to look at this, and these numbers may be hard to accurately pinpoint, we removed the number of crops in L38 in the track changed version of the revised manuscript as per the reviewer’s suggestions.

L43: You need a reference here for declines of wild pollinators. See potentially Powney, G.D., Carvell, C., Edwards, M., Morris, R.K.A., Roy, H.E., Woodcock, B.A., Isaac, N.J.B., 2019. Widespread losses of pollinating insects in Britain. Nature Communications 10, 1018.

Response: We appreciate the reviewer’s suggestions. The reference has been added in L47 in the track changed version of the revised manuscript.

L63: I think you need to specify that this could occur in a manner that is fundamentally different to what original regulatory assessments predicted for that product.

Response: We appreciate the reviewer’s suggestions. We edited L67-68 in the track changed version of the revised manuscript as per the reviewer’s suggestions.

L68: Potentially see also Milner, A.M., Boyd, I.L., 2017. Toward pesticidovigilance. Science 357, 1232-1234. Which shows a similar ting (for neonicotinoids).

Response: We appreciate the reviewer’s excellent suggestions. We feel this fits better in the conclusion of the paper and thus included it in L500 in the track changed version of the revised manuscript as per the reviewer’s suggestions.

.

L75-77: So I guess the obvious thing is to ask if ther is scope for something more formal than just a literature review – metanalysis for example. It may eb the case that current data is not expansive enough for this once you take into account the needs for some level of standardisation.   Possibly need to justify why you take this approach and not something like a meta-anlaysis.

Response: We appreciate the reviewer’s concerns. The reviewer correctly pointed to a major concern in such types of review themes. While bees can sample many chemicals in the environment, the sampling of hives as reported in the literature is far too limited to truly characterize how bees pick up chemicals or particles across time, space, crops, climates, etc. A meta-analysis would be more appropriate in settings where similar scientific approaches can be compared to each other, as in a meta-analysis of drug effects. 

L84: So given how hairy bees are relatie to other pollinating insects does this mean they are at a greater than expected risk (e.g. say compared to flies or beetles)

Response: We appreciate the reviewer’s comments. The goal of this paper is to point out gaps in knowledge, and our aim is to stimulate questions from the readers, such as this question posed by the reviewer. Bees certainly seem structurally predisposed to such exposures when compared with other insects, and even bats and birds. This exposure may also be driven by the type of particle or the crop which is sprayed, and since there are many additional factors which drive risk, we would hesitate to make this assertion.

L97: So this specifically relates to honeybees (i.e. you say bees here). I know you specify it earlier but probably need to always use honeybees. Note many other wild bees do not have corbicula on the back legs (its only really bombus and apis).

Response: We appreciate the reviewer’s observations, and clarified our language in L102 and L107 in the track changed version of the revised manuscript as per the reviewer’s suggestions. Some of the sources refer to work with bumble bees regarding electrostatic charges and flowers. We are inferring to all such published research to paint a general picture of how particles could interact with pollinators.

L105-108 (by picture) – I am guessing this is unidentified particle. Make this clear that its not necessarily a NBP.

Response: We thank the reviewer for this comment. We have inserted the word “unidentified” into the text and caption text, which can be found in in L110 and L114 in the track changed version of the revised manuscript, as per the reviewer’s suggestions.

L121: Where do you get this from?   As a back of the envelope calculation if you assume bees forage around 10 km or less from hives this is 7853 ha. If its justified this is ok, but if its of the top of your head I would be careful with this.

Response: We thank the reviewer for this comment. which is definitely true for a single bee colony. We were talking about the landscape, but the reviewer’s point of view makes more sense. Hence, we changed this in L120 in the track changed version of the revised manuscript as per the reviewer’s suggestions..

L130: also see Woodcock, B.A., Ridding, L., Freeman, S.N., Pereira, M.G., Sleep, D., Redhead, J., Aston, D., Carreck, N.L., Shore, R.F., Bullock, J.M., Heard, M.S., Pywell, R.F., 2018. Neonicotinoid residues in UK honey despite European Union moratorium. PLoS ONE 13, e0189681.

Response: We appreciate the reviewer’s suggestions. We added the reference in L129 in the track changed version of the revised manuscript as per the reviewer’s suggestions.

L157-159: I think this is a really important point – I totally agree that when it comes to applying pesticides commercial confidentiality should not be allowed.

Response: We thank the reviewer for this comment.

L172-174: I think this in some form should go in the abstract – is their space or are you limited by word count.

Response: We appreciate the reviewer’s suggestions. We carefully considered the reviewer’s comment. We do discuss size in the abstract, but the definition of nano and how it relates to NBPs is a rather long and nuanced discussion and thus refrain from detailing it in the abstract.

L177: It would be good if you could extend information on regulation (in brief) to other zones – ie..whats happening in Europe, Asia etc. Don’t need to kill yourself here, but make this more than just about the US.

Response: We appreciate the reviewer’s suggestions. We tried to avoid taking the readers down a regulatory rabbit hole. The USEPA is principally looking at nanosilver as an antimicrobial pesticide, while EFSA is looking at nanosilver in cosmetics. We did include a little information about how EFSA is contemplating what “nanopesticide” even means, please see changes in L188-L197 in the track changed version of the revised manuscript for the edited details.

L190-191: Again I think this is such an important point, the lack of clarity on how formulations including NBPs should be treated from a regulatory perspective given that while not an active ingredient they fundamentally alter properties of the pesticides. 

Response: We appreciate the reviewer’s comments. This is an excellent restatement of the point we are trying to make.

L222 Great figure. One that would be very cool would be to have the same pesticide but for different formulations (I am not clear if this figure may have this a t least in some of the panes).

Response: We appreciate the reviewer’s suggestions. That is a very interesting idea, to see how many formulations we can look at with the same active ingredient. However, this is beyond the scope of the present review manuscript.

L235-255: I wonder whether some of this detail could go in an appendix, to help with the flow of the paper. Maybe, not sure on this.

Response: We appreciate the reviewer’s suggestions. However it would be inconvenient for a reader to have to go back and forth between an appendix to the figure which the text refers to. Hence, we would request to keep the information as it is.

L256: I find this deeply worrying

Response: We appreciate the reviewer’s concerns. The present review paper is talking about bees, and this may potentially have ramifications for human health as well.

L300-303: Should this definition not go earlier on in the review?

Response: We appreciate the reviewer’s suggestions. As we mentioned before, the definition of nano vs what is a nanopesticide is a kind of nuanced conversation. We tried to build up to it, and explain what is being regulated first. The definition would include some things that aren’t currently regulated.

L310: mentioned this earlier but your scope seems to vary. Is this focused on honeybees specifically, bees in general or pollinators (which can be a much wider taxonomic group to include for example flies and butterflies). The reason I think you could perhaps focus on bees is your emphasis on electrostatic forces and how hairs exasperate these. Certainly this would seem to be relevant for most bees (note while they don’t all have corbicula, they do tend to have pollen storage structures of one type or another), but may be less pertinent for say anthomyiidae flies (note you may want to consider some hoverflies as due to their mimicry they can be hairy).

Response: We appreciate the reviewer’s concerns. The scope varies because honey bees and bumble bees are well studied and are the focal examples. However, this could definitely apply to all pollinators. However, there is an extensive gap in knowledge regarding this. NBPs are meant to stick to pest insects, and are likely designed to do so purposely. We may potentially look into what types of particles flies and other insects pick up. However, that is beyond the scope of the present review.

L320-321: Add a reference for this – it’s a lot of detail in the paragraph, best to add some citations. E.g. Cresswell, J.E., 2011. A meta-analysis of experiments testing the effects of a neonicotinoid insecticide (imidacloprid) on honey bees. Ecotoxicology 20, 149-157. But frankly there billions of NI references to pick from.

Response: We appreciate the reviewer’s comments. Our point here relates to fugitive dust, while the paper the reviewer cites mostly relates to laboratory experiments. We have added another reference to expand on the dust issue: Planting of neonicotinoid‐treated maize poses risks for honey bees and other non‐target organisms over a wide area without consistent crop yield benefit by Krupke et al 2017. Please see L338 in the track changed version of the revised manuscript.

L333: Again see PLoS ONE 13, e0189681.

Response: We thank the reviewer for this suggestion. We have included this citation in L341 in the track changed version of the revised manuscript as per the reviewer’s suggestions.

L348-357: I wonder whether this warrants a separate section – its short I know but its focus on non-pesticide risks associated with nano-particles is distinct in a way.

Response: We appreciate the reviewer’s suggestions. This section is focused on showing that both intentionally and unintentionally created particles and nanoparticles, some of which are associated with pesticides, have posed risks in the past.

L371: So these tend to be orientate towards honeybees, other wild bee species can have longer flight periods throughout the day and as such may still be at risk even where these spraying time precautions are applied.

Response: We appreciate the reviewer’s suggestions. This is an excellent comment, but we are trying not to diverge greatly from discussing NBPs.

411: So one thing that may be relevant (sorry if you consider it later on) is the effect of the part of NBPs on the persistence of the pesticides activity – I am thinking here of sub lethal effects most widely linked with neonicotinoids as seed treatments and how long term low level exposure risks took a long time to be appreciated. If this changes dramatically with the use of NBPs this could create new problems for existing pesticides. This is especially true given the problems with assessing LC50s under lab conditions – i.e. the control bees tend to die quickly so even long term chronic exposure assessments rarely extend beyond 10-12 days.

Response: We appreciate the reviewer’s suggestions. This has already been referred to in the manuscript.

L430: Another potentials consideration is if NBPs alter solubility in oils / water – i.e. could end up in wax where this was not an issue before – or vice versa

Response: We appreciate the reviewer’s suggestions. This has already been referred to in the manuscript.

L449 – ok great you do cover this point.

Response: We appreciate the reviewer’s time in carefully reviewing our manuscript.

Reviewer 2 Report

The paper provides a comprehensive overview of the physics and chemistry of nanotechnology embedded pesticides and the potential impact of these formulations on pollinator insects. Key mechanisms that are likely to be altered by structure sizes, such as the uptake during pollination or a subsequent accumulation, are well summarized and referenced. As there are no experimental observations of effects of nanotechnology-based pesticides (NBPs) yet, the paper formulates risks and proposes studies to be performed. In addition to a comprehensive literature review, the work contains some original electron microscopic data on the size and structure of some commercial insecticide products.

The paper is a very well-structured and well-presented summary of two areas that have recently started to overlap. It will be useful for both scientific communities, the developers of NBPs and the chemical ecologists who will be conducting future studies on their risk to pollinators. I, therefore, recommend this paper for publication.

The only inadequacy I see in this work is an inaccurate description of the status quo of NBP commercialization: I miss a clear overview of what are the products on the market that have been identified as such and what are their characteristics. The effort in the form of the electron microscope images of some commercial formulations (Figure 4) does not allow a clear assessment of whether the objects shown are actually nanotechnologically manufactured. While this is clearly the case in Fig. 4J, showing spheres of uniform submicron size, many other images do not show such a clear scenario, and structures could well be arbitrary recrystallizations of classical pesticide formulations.

For this reason, I would ask to add the following information, ideally in the form of a table to provide an overview:

-Which of the producers claim their products are nanotechnology-based formulations?

-Is any product actually labeled as "Nano" as required e.g. by the EU regulation for biocides <100nm?

-Which of the structures in Fig. 4 fall doubtlessly under what is stated in the papers as Kah’s definition of Nanopesticides?

-Please provide also images of larger samples sizes that give an idea of the overall distribution of particle sizes and shapes (possibly with the current zooms as inlets)

-Add to the table an estimate of average particle sizes and their variances within your samples

This would provide important information for the community to choose potential products for future studies on pollinators.

Finally, I would suggest highlighting explicitly not only the potentially increased toxicity of NBP formulations but their potential to induce fatal sublethal effects on the behavior of pollinators, which set in at concentrations far below toxicity thresholds (for recent review e.g. Cabirol et al. Insects 2019).

Minor points:

-Table 1: remove the nanopesticide lines from Table 1, as there isn't any information on the effects on bees available yet, nor on their physical and chemical properties (otherwise mention and reference those)

-Line 206: “transfer potential of these particulates to off-target species”: In [45] I didn’t find such studies.

-Line 300: If what you cite here is later referred to as “Kah’s definition”, you should make this clear at this point.

-Line 318: “Current encapsulated NBP pesticide formulations are orders of magnitude smaller in particle size (Table 1, Figures 4 and 5)”. This suggests a reduction of at least a factor of 100 in size. Is this fact supported by the structures you show in Fig. 4 which are mostly >1 micron?

-Line 404: “NBPs may reach target sites at higher doses compared to conventional formulations [127]”: the referenced paper provides no additional information, but repeats the same hypothesis.

Author Response

The paper provides a comprehensive overview of the physics and chemistry of nanotechnology embedded pesticides and the potential impact of these formulations on pollinator insects. Key mechanisms that are likely to be altered by structure sizes, such as the uptake during pollination or a subsequent accumulation, are well summarized and referenced. As there are no experimental observations of effects of nanotechnology-based pesticides (NBPs) yet, the paper formulates risks and proposes studies to be performed. In addition to a comprehensive literature review, the work contains some original electron microscopic data on the size and structure of some commercial insecticide products. The paper is a very well-structured and well-presented summary of two areas that have recently started to overlap. It will be useful for both scientific communities, the developers of NBPs and the chemical ecologists who will be conducting future studies on their risk to pollinators. I, therefore, recommend this paper for publication.

The only inadequacy I see in this work is an inaccurate description of the status quo of NBP commercialization: I miss a clear overview of what are the products on the market that have been identified as such and what are their characteristics. The effort in the form of the electron microscope images of some commercial formulations (Figure 4) does not allow a clear assessment of whether the objects shown are actually nanotechnologically manufactured. While this is clearly the case in Fig. 4J, showing spheres of uniform submicron size, many other images do not show such a clear scenario, and structures could well be arbitrary recrystallizations of classical pesticide formulations.

Response: We thank the reviewer for carefully reading our manuscript and for the reviewer’s thoughtful comments. We considered each suggestion very carefully. Many of the reviewer’s observations and questions relate to the gaps in knowledge that we are trying to highlight in this manuscript. We are gratified that the review of our paper reflects the important questions we hope readers will ask.

Unfortunately, pesticide manufacturers are not required to reveal most product ingredients other than the active ingredients which we have noted in this paper, and it would be impossible to know which pesticides are formulated using particles without looking. Although much of formulation chemistry is confidential business information (see L158 in the revised manuscript), looking through the literature on pesticide formulation will lead readers to images of particles which are comparable to our photographs. We compared a few of the formulations to the active ingredient alone and found them to be different from the particles illustrated here, although we did not collect a consistent series of photographs which could be used as a figure. Even if we were mistaken, and any of these formulations resulted in “arbitrary recrystallizations of classical pesticide formulations”, this could result in inadvertently created nanoparticles, and the same open questions about movement in the environment and bee exposure would apply.

For this reason, I would ask to add the following information, ideally in the form of a table to provide an overview:

-Which of the producers claim their products are nanotechnology-based formulations?

-Is any product actually labeled as "Nano" as required e.g. by the EU regulation for biocides <100nm?

Response: We appreciate the reviewer’s suggestions. Some fraction of pesticides can dip below the 100 nm as shown in https://www.ncbi.nlm.nih.gov/pubmed/26540086. The features of many pesticides may be below 100 nm, such as the width of pores or sheet-like structures in clays. Many of those are below 1000nm, up to several microns. To the best of our knowledge, we are not aware if any pesticide is currently labeled with the nano term.

Part of the EU definition for biocidal nanomaterials is materials having 50% or more of the particles with a size of 1-100 nm in at least one dimension, and in our manuscript this only applies to nanosilver of all the materials discussed. This is an issue we are trying to highlight – only the smallest particles get regulated. If somewhat larger particles such as NBPs have size-related properties that affect risk to bees, humans, or other organisms, they should also receive regulation.  We added a small amount of text relating to EU regulation of nanopesticides in L188-L197 in the track changed version of the revised manuscript. When there is more data regarding the risk or lack of risk from nanopesticides, we hope that regulatory agencies discuss this issue in detail. At this time, there is little to report.  

Which of the structures in Fig. 4 fall doubtlessly under what is stated in the papers as Kah’s definition of Nanopesticides?

Response: We appreciate the reviewer’s concerns. As per Kah’s definition:

Size up to 1000nm – many of the particles and features of particles fit this characteristic.

Contains the prefix nano – no, nano would be an alarming word to the public when associated with a pesticide. We think that the pesticide industry would want to avoid that. It is also possible that industry formulators think of these particles as colloids, because they are in solution until they are applied in the environment.

or –

Makes claims based on size – For most of these products, the size of the particles enables the hydrophobic pesticide to stay in an aqueous solution. It is likely that the particles provide other properties such as extended residual toxicity etc., see Kah’s reference: https://www.sciencedirect.com/science/article/pii/S0160412013002754. To reiterate, pesticide companies hold their formulations close to their chests as confidential business information. Our related hypothesis is that particle properties may also influence risk, and that this is certainly a research gap that needs further studies in the future. Many of Kah’s articles refer to the same kinds of structures we are discussing in this paper, and the studies have broadly termed them as nanopesticides.

Please provide also images of larger samples sizes that give an idea of the overall distribution of particle sizes and shapes (possibly with the current zooms as inlets). Add to the table an estimate of average particle sizes and their variances within your samples. This would provide important information for the community to choose potential products for future studies on pollinators.

Response: We appreciate the reviewer’s suggestions. We have been asked to return this manuscript within 5 days, and are unable to produce additional SEM photographs or analysis in that time. We have already taken the very unusual step of including original SEM photos in a review paper. We did so because we have frequently encountered incredulity (maybe it is just arbitrary recrystallizations of classical pesticide formulations, as above). There are many journal articles describing nanopesticides. In addition to size ranges, we very much hope we and other researchers will continue to examine the interactions between pollinators and these particles, perhaps using SEM.

Finally, I would suggest highlighting explicitly not only the potentially increased toxicity of NBP formulations but their potential to induce fatal sublethal effects on the behavior of pollinators, which set in at concentrations far below toxicity thresholds (for recent review e.g. Cabirol et al. Insects 2019).

Response: We appreciate the reviewer’s suggestions. Our manuscript is less focused on toxicity and more on the exposure. If exposure to the active ingredient is altered due to being attached to a particle, we hope future studies will examine any added risk of a variety of toxicological responses. Cabirol et al. discusses mechanism of action of neonicotinoids. We do not cite mechanisms of action for other toxins in this paper, and so did not cite this one https://www.mdpi.com/2075-4450/10/10/344/htm.

Minor points:

Table 1: remove the nanopesticide lines from Table 1, as there isn't any information on the effects on bees available yet, nor on their physical and chemical properties (otherwise mention and reference those)

Response: We appreciate the reviewer’s comments. The intention of this paper is to highlight that in the context of effects of similarly sized particles, there is no information on pesticides associated with particles.  These lines are meant to help illuminate that information gap.

Line 206: “transfer potential of these particulates to off-target species”: In [45] I didn’t find such studies.

Response: We appreciate the reviewer’s suggestions and thank the reviewer for pointing this. We have now removed this reference in the revised manuscript.

Line 300: If what you cite here is later referred to as “Kah’s definition”, you should make this clear at this point.

Response: We thank the reviewer for this excellent point. We have now modified this sentence, please see our changes in L307 in the track changed version of the revised manuscript as per the reviewer’s suggestions.

Line 318: “Current encapsulated NBP pesticide formulations are orders of magnitude smaller in particle size (Table 1, Figures 4 and 5)”. This suggests a reduction of at least a factor of 100 in size. Is this fact supported by the structures you show in Fig. 4 which are mostly >1 micron?

Response: We appreciate the reviewer’s comments. Sizes of encapsulated pesticides have been observed down to 200 nm (https://www.ncbi.nlm.nih.gov/pubmed/26540086), which compared to 50 um for PennCap M, is greater than a factor of 100 difference.

Line 404: “NBPs may reach target sites at higher doses compared to conventional formulations [127]”: the referenced paper provides no additional information, but repeats the same hypothesis.

Response: We appreciate the reviewer’s suggestions. Nanopesticides may be targeted, slow release or other forms and are thus designed to reach the target at higher effective doses. We appreciate the reviewer’s suggestions and agree that this should be researched further in the future.